# First description of Nodding Syndrome in the Central African Republic

Salvatore Metanmo[1,2,3], Farid Boumédiène[1,2,3], Pierre-Marie Preux[1,2,3], Robert Colebunders[4], Joseph N. Siewe Fodjo[4,5], Eric de Smet[4], Emmanuel Yangatimbi[6], Andrea S. Winkler[7,8], Pascal Mbelesso[6], Daniel Ajzenberg[1,2,3]*

1 Institut national de la santé et de la recherche médicale (INSERM), U1094, Tropical Neuroepidemiology, Limoges, France, 2 Univ. Limoges, U1094, Tropical Neuroepidemiology, Institute of Epidemiology and Tropical Neurology, Limoges, France, 3 Institut de recherche pour Le développement (IRD), Associated Unit, Tropical Neuroepidemiology, Limoges, France, 4 Global Health Institute, University of Antwerp, Antwerp, Belgium, 5 HILPharma Health Organization, Yaoundé, Cameroon, 6 Faculty of Health Sciences, University of Bangui, Bangui, Central African Republic, 7 Department of Neurology and Center for Global Health, Klinikum rechts der Isar, Technical University of Munich (TUM), Munich, Germany, 8 Centre for Global Health, Institute of Health and Society, University of Oslo, Oslo, Norway

* daniel.ajzenberg@unilim.fr

**Data Availability Statement:** All relevant data are within the manuscript and its Supporting Information files.

## Abstract

### Background

The term Nodding Syndrome (NS) refers to an atypical and severe form of childhood epilepsy characterized by a repetitive head nodding (HN). The disease has been for a long time limited to East Africa, and the cause is still unknown. The objective of this study was to confirm the existence of NS cases in Central African Republic (CAR).

### Methodology/Principal findings

This was a cross-sectional descriptive study in the general population. The identification of NS cases was conducted through a door-to-door survey in a village near Bangui along the Ubangui River. Based on Winkler's 2008 and the World Health Organization (WHO)'s 2012 classifications, the confirmation of cases was done by a neurologist who also performed the electroencephalograms. No laboratory tests were done during this investigation. Treatment was offered to all patients. A total of 6,175 individuals was surveyed in 799 households. After reviewing the cases, we identified 5 NS cases in girls aged between 8 and 16. The age of onset of the seizures was between 5 and 12 years of age. Two cases were classified as "HN plus" according to Winkler's 2008 classification. Four NS cases were classified as probable and one as confirmed according to the WHO's 2012 classification. Three of them presented with developmental delay and cognitive decline, and one had an abnormally low height-for-age z-score. Electroencephalographic abnormalities were found in four patients.

### Conclusions/Significance

Nodding Syndrome cases were described in CAR for the first time. Despite certain peculiarities, these cases are similar to those described elsewhere. Given that only a small part of

**Funding:** The author(s) received no specific funding for this work.

**Competing interests:** The authors have declared that no competing interests exist.

the affected area was investigated, the study area along the Ubangui River needs to be expanded in order to investigate the association between *Onchocerca volvulus* and NS and also evaluate the real burden of NS in CAR.

## Author summary

Nodding Syndrome (NS) is a form of severe epilepsy that affects children in Africa. Thousands of children have been affected since its first description 60 years ago in East Africa, particularly Tanzania, South Sudan, and Uganda. Its evolution is marked by the appearance of many serious complications such as stunting, wasting, delayed sexual development, and psychiatric illness that can lead to death in some cases. Both the future of the affected children and the present of the adults in charge of them are hampered by an intolerable level of social and economic harm. Here, we describe new cases of NS that emerged in a context of extreme poverty in the Central African Republic (CAR). Because the area at risk of NS was partially investigated in this study, we urge to carry out a larger-scale study in order to assess the real burden of NS in CAR and draw the attention of public authorities.

## Background

The term «Nodding Disease» appeared for the first time in 2003 referring to an atypical form of epilepsy presenting a repetitive « head nodding (HN) » triggered by food or cold temperatures, in children aged 3–18 years [1]. It was in 2012 that the expression "nodding syndrome (NS)" appeared for the first time at a World Health Organization (WHO) conference [2]. Initially the disease seemed restricted to East Africa since confirmed NS cases were retrospectively reported in Tanzania in the 1960s [3], and later in South Sudan and northern Uganda in 1990 and 2013, respectively [4]. Since then, confirmed cases were reported in Cameroon (Central Africa) and cases were suspected in Liberia (West Africa) and the Democratic Republic of Congo (DRC, Central Africa) [5–7]. It is estimated that thousands of children have been affected in Uganda, Tanzania, and Sudan [8].

Involuntary and repetitive head nodding (i.e. forward bobbing of the head) is a pathognomonic sign of NS. Other manifestations which may complete this clinical picture include: generalized seizures, cognitive impairment, psychiatric symptoms, growth retardation, and morphological malformations; the evolution of these symptoms often results in significant disability and sometimes death [8–10]. Moreover, abnormalities on electroencephalography (EEG) have been observed in some NS cases [11–13]. For example, EEG was abnormal or even markedly abnormal, showing a typical 2.5 spike wave pattern when recorded immediately after a nodding episode in two cases and electrical depression during nodding episodes in two other children [13].

There are two classifications of NS. The first was proposed by Winkler et al in 2008 and classifies patients in two categories: one with "HN alone" and one with "HN plus" i.e. HN associated with other forms of epilepsy [14]. The second classification was proposed following the first international conference on NS organized by the WHO in 2012 in Kampala. It classifies cases as suspected, probable and confirmed [2]. On the basis of new findings and the limitations of the aforementioned classifications, an expert consensus suggested a refined and

improved version at an NS conference in Gulu in 2015, but this new classification has not yet been universally adopted [15].

Regular use of routine anti-epileptic drugs (AED) may be efficient in controlling the seizures in NS patients [16]. Even though the cause of this syndrome remains unclear, an association was found in several areas with onchocerciasis or river blindness due to the parasite *Onchocerca volvulus* [17,18]. Little data is available on epilepsy in CAR, particularly in this area. The most recent data show a prevalence of epilepsy of 2.8‰ in Bangui [19]. In the First International Workshop on Epilepsy Associated with Onchocerciasis held in Antwerp (Belgium) from 12 to 14 October 2017 [20], we were told informally by colleagues from Central African Republic (CAR) that they suspected NS cases near Bangui. We conducted this study to confirm these anecdotal reports, and provide detailed clinical descriptions of NS cases in this area.

## Methods

### Ethics statement

Prior to conducting the study, ethical approval was obtained from the CAR Scientific Committee (authorization number: N˚_32_UB/FACSS/CSCVPER/19). Written Formal consent was obtained from the parent/guardian in each case.

### Study setting

Fieldwork was carried out between November 23rd and December 8th, 2019, in the village of Landja (**Fig 1**). The **Fig 1** also shows the countries where cases of NS have been suspected and/ or confirmed [5–7]. Landja is a rural area of a little more than 37,000 inhabitants located 15 kilometers at the northeast of Bangui, along the Ubangi River where active onchocerciasis transmission was recently confirmed [21]. The inhabitants live mainly from fishing and agriculture and they speak Sango, the local national language, and a few of them, French. The village is divided into 6 districts, the first being the closest to Bangui and the easiest to access. In this district, there is a health center belonging to the Bel-Espoir non-governmental organization (NGO) and managed by a state-registered nurse. This health center provides first aid and minor surgery. A second NGO called Fracarita is active in the area and its main role is the monthly free distribution of AED such as valproic acid and phenobarbital in this area.

### Study design and procedures

This was a cross-sectional community-based study. The investigators were organized in three teams, each team consisting of a final-year medical student from the"Hôpital de l'Amitié" in Bangui and a local village resident who had been involved in previous surveys. NS cases were identified using an active case detection approach via door-to-door visits in the first district of Landja.

This district was chosen for two reasons: firstly, it was in this district that the Fracarita focal point initially suspected HN cases among children and issued the alert; secondly, for logistical reasons in particular the easy access to the zone. Of note, no data on *Onchocerca volvulus* endemicity were available in this area at that time. All the households were investigated and after presenting the study objectives, informed consent was obtained from the household head or his representative. Screening of epilepsy and NS cases was performed with an eight-question screening questionnaire (S1 File), which was designed for the survey and pre-tested in healthy volunteers but without performance validation. This screening questionnaire was submitted to all household members by final-year medical students. Detected NS/epilepsy cases were referred to the village health center, where a trained neurologist (PM) performed a detailed

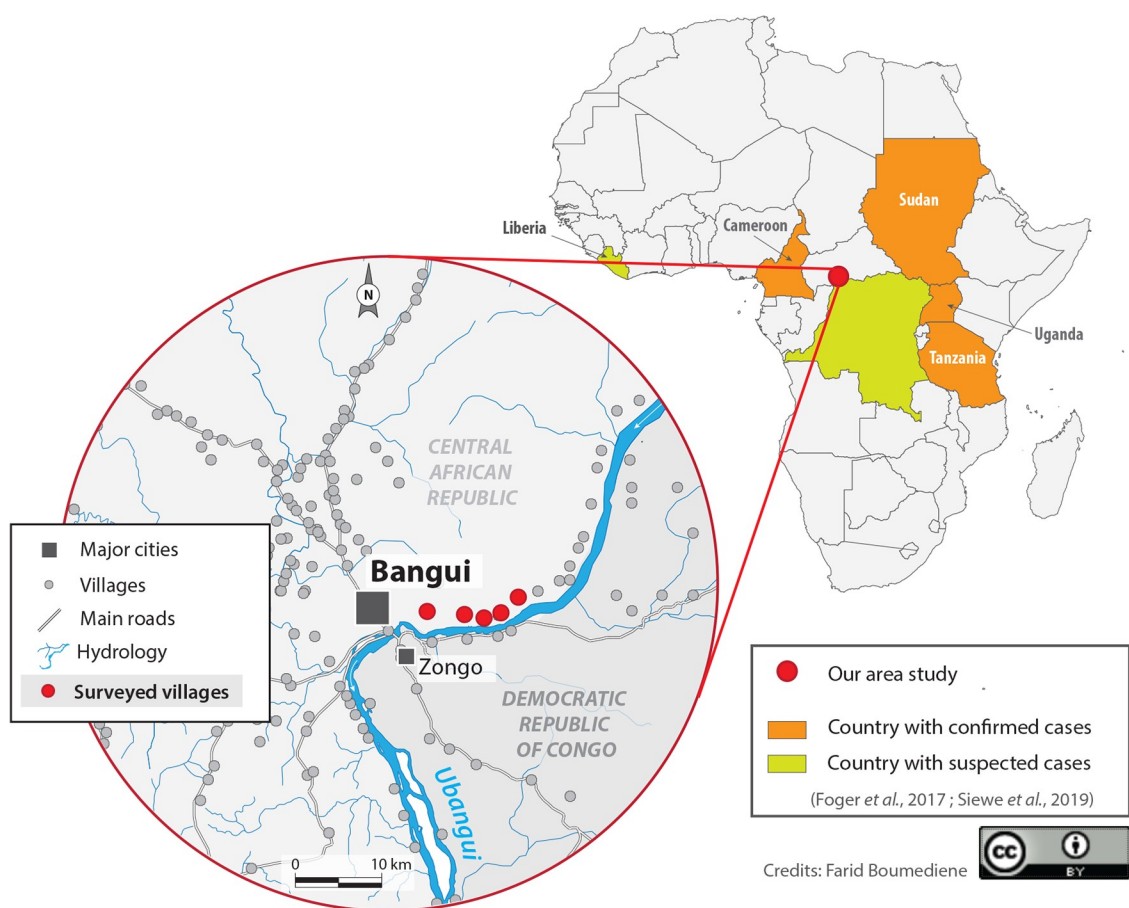

**Fig 1. Map of Africa with confirmed and suspected cases of Nodding Syndrome and study area; the whole map was created by Dr Farid Boumédiène.**

clinical and neurological examination to ascertain the diagnosis. Confirmation of epilepsy cases was supported by the collection of additional information about socio-demographic characteristics, anthropometric data, and seizure history using a second pre-tested questionnaire (S2 File). The confirmation of NS cases was done through a third questionnaire via the Kobocollect tool available in electronic devices (S3 File). A fourth questionnaire (S4 File) was used by a Medical Doctor (SM) to obtain additional data from confirmed NS cases regarding the onset and natural history of NS, including antenatal and childhood history, psychomotor development, medical and surgical history, history of medication use, and symptoms experienced before/after the onset of nodding seizures.

Persons with confirmed clinical NS agreed to undergo electroencephalogram (EEG) ("DELL de Natus Medical Incorporated, DBA Excel-Tech LDT. (XLTEK). 2568 Bristol Circle, Ontario, L6H5S1 Canada") performed by the neurologist (PM) from Bangui. Once diagnosed with NS, the patients received anti-epileptic treatment, and were invited to attend monthly or quarterly follow-up visits with a neurologist.

## Classifications

Epilepsy was defined as the occurrence of two or more unprovoked seizures at least 24h apart, as recommended by the International League Against Epilepsy [22]. As for NS, we adopted the

classifications proposed by the World Health Organization (WHO) for the differentiation of suspected, probable, and confirmed NS cases [2]. We also used the NS classification proposed by Winkler et al.: "HN only" (exclusively head nodding) vs. "HN plus" (head nodding associated with other seizure types) [14].

Retarded growth was defined as a height-for-age index less than the fifth percentile, based on the Centers for Disease Control and Prevention (Atlanta, USA) growth curves for children aged 2–20 years [23]. The weight-for-age index was also obtained from these curves (Z-score and percentile).

Developmental delay and cognitive decline was classified into three stages (mild, moderate and severe) based on clinical criteria according to the WHO [24].

### Statistical analysis

The Kobocollect tool (Harvard Humanitarian Initiative and Cambridge, USA; https://www.kobotoolbox.org/) used for investigating NS cases was installed on a tablet that allowed the data to be entered offline and then sent to the server of the University of Limoges. Data security was in agreement with current French medical data confidentiality standards. Data were analyzed using Excel 2016. We report descriptive findings with proportions as percentages. No inferential statistical analyses were performed.

## Results

A total of 6,175 individuals were screened in 799 households of the study area, of whom 75 were examined by the neurologist, 55 were diagnosed as having epilepsy corresponding to a prevalence of 8.9 per 1000. Seven cases were initially identified as having NS, but after reviewing the cases, we excluded two cases with HN: one had a history of tubercular meningitis at the age of four years followed by HN onset seven years later; the other had experienced several episodes of severe acute malnutrition by the age of six months, and HN started at age five (see more details in S5 File). The five NS cases corresponded to a prevalence of 0.8 per 1000 for NS and accounted for 8.3% of all persons with epilepsy (PWE) in our study population. In this paper, the focus is on describing the NS cases only.

### Clinical description of Nodding Syndrome cases

**Case N° 1.**   Female, aged 11 years, who has never been at school. Hypertonic convulsive seizures reportedly started at 30 months of age, and HN began at the age of six. Attention deficits were noticed by her mother 32 months prior to HN onset. About 24 months after the onset of NS, other symptoms emerged including difficulties to concentrate, hyperactivity, memory deficit-s, wandering behavior and four-limb deformities.

At the clinic, the patient presented with a behavioral disorder, agitation, aphasia, left spastic hemiparesis, left inwardly turned foot, tendon retractions of the two lower limbs more severe on the left and stumbling gait. Her memory and language were also impaired. We equally noted moderate developmental delay and cognitive decline, retarded growth, and stiffness of the joints and limbs. Finally, this patient was bulimic but with a normal weight-for-age index (Z-score = -1.19, percentile = 11.7%). This is the only case in which a HN was witnessed by the research team; the seizure was triggered by noise, lasted for about 30 seconds, and consisted of about 8 HN/min.

**Case N° 2.**   Female, aged eight years, with pronounced learning difficulties at school, which got worse after HN started. She reportedly experienced attention deficits shortly before HN onset, but the exact timing could not be specified. HN started at the age of five and persisted to date. The main HN trigger was cold. The patient was mostly lucid but often had

aggressive episodes, reported to have started three months after HN onset. Memory impairment, visual and auditory hallucinations, began 24 months after HN onset. Prolonged insomnia and saddened mood were noted one and four months after HN started, respectively. The rest of the physical examination was unremarkable.

**Case N˚ 3.** Female, aged 16 years, whose HN started at the age of eight. Although this patient was an average school student, she had dropped out of school after HN onset. About 12 months following HN onset, she started experiencing learning difficulties and memory deficits. Auditory hallucinations started 24 months after HN onset. Prolonged mood changes, including episodes of sadness, also developed but the timing was unclear. A few months after the onset of HN, generalized myoclonic seizures started and were still present at the time of this study.

The frequency of HN could not be determined, and no specific HN trigger was identified. The neurological examination found a monoparesis with amyotrophy and a right claw hand, a right hand drop, and spastic, stumbling gait.

**Case N˚ 4.** Female, aged eight years, who reported onset of HN 11 months before this study, which led to her dropping out of school. Already at age five, a number of signs had appeared and were still present during our neurological examination: reduced concentration, blank starring episodes and excessive sleep. HN started in 2019 with an estimated frequency of 11–15 per minute, and was often triggered by meals taken during cold weather. Learning difficulties, little eating, lower than expected weight for age (Z-score = -3.60, percentile = 0.1%), and other symptoms appeared almost simultaneously with HN. Overall, this patient was ill-looking with obvious growth retardation (Z-score = -2.12) and loss of interest in activities. The rest of the clinical examination was unremarkable.

**Case N˚ 5.** Female, aged 15 years, unschooled, and with a family history of epilepsy (brother). At the age of 12, nodding seizures started with a frequency of 5 HN/min, which was the same seizure frequency reported at the time of the survey. There were no specific triggers for the HN. Excessive somnolence was noted by her mother prior to HN onset, with worsening thereafter. Twelve months after HN onset, she experienced progressive weakness of both lower limbs and the right upper limb which evolved into a spastic syndrome, resulting in an unstable gait confirmed on clinical examination.

## Synthesis of the main characteristics of NS cases in this study

All NS cases were girls, without family ties, and their ages ranged from eight to 16 years (**Table 1**). Three of them (71%) were born in Ouango town. The age of HN onset was between five and 12 years, and the most recent case was in 2019. Two cases were classified as "HN plus" according to the classification by Winkler et al. (2008) and reported that generalized seizures preceded the onset of HN. Only one case (case 1) fully satisfied the criteria for « confirmed » NS according to the WHO classification. **Table 2** summarizes the details of the WHO classification for each NS case.

The frequency of HN varied from 5 HN/min to 11–15 HN/min. Two cases presented with moderate developmental delay and cognitive decline and one case with mild developmental delay and cognitive decline (not interfering with daily activities). Two patients had a low height-for-age. Several seizure triggers were reported, including noise (in one out of five cases) and cold (in two out of five cases).

The medical history of the five NS cases is presented in **Table 3**. All reported uneventful pregnancies and childbirth. Based on the Tanner classification for sexual development [25], none of the three cases who reached puberty had a delayed development of secondary sexual characteristics.

**Table 1. Socio-demographic and clinical characteristics of Patients with Nodding Syndrome, Central African Republic, 2019.**

| Characteristics | 1 | 2 | 3 | 4 | 5 |
|---|---|---|---|---|---|
| Age (years) | 11 | 8 | 16 | 8 | 15 |
| Sex | Female | Female | Female | Female | Female |
| Place of Birth | Ouango | Mbaïki | Ouango | Ouango | Gboko |
| Age at HN[a] onset (year) | 8 (2014) | 5 (2016) | 8 (2011) | 8 (2019) | 12 (2016) |
| Other epilepsy | Generalized tonic-clonic | No epilepsy | Generalized myoclonic | No epilepsy | No epilepsy |
| Winkler's classification | HN plus | HN only | HN plus | HN only | HN only |
| WHO[b]'s classification | Confirmed | Probable | Probable | Probable | Probable |
| Frequency (HN/min) | 8 | 6–10 | NA | 11–15 | 5 |
| Trigger factor | Noise | Cold | Noise, heat | None | Meal, cold |
| Neurological examination | Abnormal/focal | Abnormal | Abnormal/focal | Abnormal | Abnormal/focal |
| Developmental delay and cognitive decline | Moderate | Mild | No | Moderate | No |
| Growth retardation | Yes | No | No | Yes | No |

[a]HN, head nodding.

[b]WHO, World Health Organization.

Only one patient (case 1) was treated with AED (phenobarbital) since the onset of symptoms at the age of 2.5 years. We could not have precise information but it seems that she took 100mg/day of phenobarbital irregularly based on drug availability in the NGO Fracarita. Moreover, only case 5 had a family history of epilepsy. Three cases had experienced food shortages and two had lived in camps during the war.

EEG was normal in one case (Case 5). Overall, the basic rhythm varied between 8 and 10 hertz and was reactive with eye opening. In one patient (case 1), the basic rhythm showed

**Table 2. Case classification of Nodding Syndrome in Central African Republic, 2019.**

| Case classification* | Case Number | | | | |
|---|---|---|---|---|---|
| | 1 | 2 | 3 | 4 | 5 |
| **Suspected case**: previously healthy person who was reported nodding their head | + | + | + | + | + |
| **Probable case**: two following major criteria and at least one minor criterion | | | | | |
| **Major criteria** | | | | | |
| Age 3–18 years at the onset of head nodding | + | + | + | + | + |
| Nodding frequency 5–20 times/minute | + | + | NA[a] | + | + |
| **Minor criteria** | | | | | |
| Other neurological abnormalities (cognitive decline, dropping out of school due to cognitive/behavioural problems, other seizures or neurological abnormalities) | + | + | + | + | - |
| Clustering in space and time with similar cases | + | + | + | + | - |
| Triggered by eating or cold water | - | + | - | + | - |
| Delayed or stunted growth | + | - | - | + | - |
| Delayed sexual or physical development | - | - | - | - | - |
| Psychiatric symptoms | + | + | + | - | - |
| **Confirmed case**: probable case with documented head nodding episodes | | | | | |
| Observed and recorded by a trained health-care worker, or | + | - | - | - | - |
| Videotaped head nodding episode or | - | - | - | - | - |
| Video/electroencephalogram/electromyogram documenting head nodding as atonic seizure | - | - | - | - | - |

*Case classification of Nodding Syndrome proposed by the World Health Organization in 2012 in Kampala

[a]NA, not available; +, present; -, absent.

**Table 3. History of patients with Nodding Syndrome, Central African Republic, 2019.**

| Background | Case Number | | | | |
|---|---|---|---|---|---|
| | **1** | **2** | **3** | **4** | **5** |
| Place of birth: health center/hospital | + | + | + | + | - |
| Normal childbirth | + | + | + | + | + |
| Birth trauma | - | - | - | - | - |
| Prematurity | - | - | - | - | - |
| Cry at birth | + | + | + | + | + |
| Psychomotor development | - | + | + | + | + |
| Sexual development | + | NA[a] | + | NA | + |
| Went to school | - | + | + | + | - |
| Pathology before head nodding | - | - | - | - | - |
| Febrile convulsions in childhood | - | - | - | - | - |
| Use of antiepileptic drugs | + | - | - | - | - |
| Use of traditional medicine | - | - | - | - | - |
| Family history of epilepsy and/or head nodding | - | - | - | - | + |
| Twin | - | - | - | - | - |
| Food shortages | - | + | + | - | + |
| Refugee Camp | - | + | + | - | - |
| Bulimia | + | - | - | - | - |

[a]NA, not available; +, present; -, absent.

theta waves of 6 hertz in the inter-ictal phase, and during somnolence we recorded diffuse slow multiple spikes occurring in a paroxysmal way and predominantly in the left hemisphere (**Fig 2**). Photostimulation and hyperventilation were without effect. In this case, a nodding seizure episode was triggered by noise during EEG recording, without any noticeable alterations in the wave pattern. In case 2, the EEG patterns showed diffuse slowing of cerebral activity during body movements associated with generalized slow spikes. Hemispheric depression was observed in case 3 with no critical abnormalities. In case 4, the basic rhythm in the alpha frequency band of 8 hertz was marked by the occasional appearance of slow diffuse theta waves of 4/5 hertz; hyperventilation led to a diffuse slowing of the waves, and photostimulation did not cause any change (**Fig 3**).

## Discussion

Based on our results, the main discussion topic is to debate whether these first cases described in CAR are really NS cases or an epilepsy syndrome of which HN could be part, but does not necessarily represent NS. Indeed, some findings in the history of NS, the clinical and even paraclinical presentation can be confusing. We will present the pros and the cons related to the diagnosis of NS in our study.

### The pros

**Socio-demographic characteristics and background.** The age of onset of HN ranged from five to 12 years, falling within the WHO probable case definition [2] and similar to what had been reported by several other authors [6,11–14,26]. This would correspond to an appearance of cases in CAR between 2011 and 2019 suggesting a recent and probably still incidental event. Families had a low socio-economic status like the majority of NS cases described both in East Africa since its discovery [3,13,18] and in Cameroon [6].

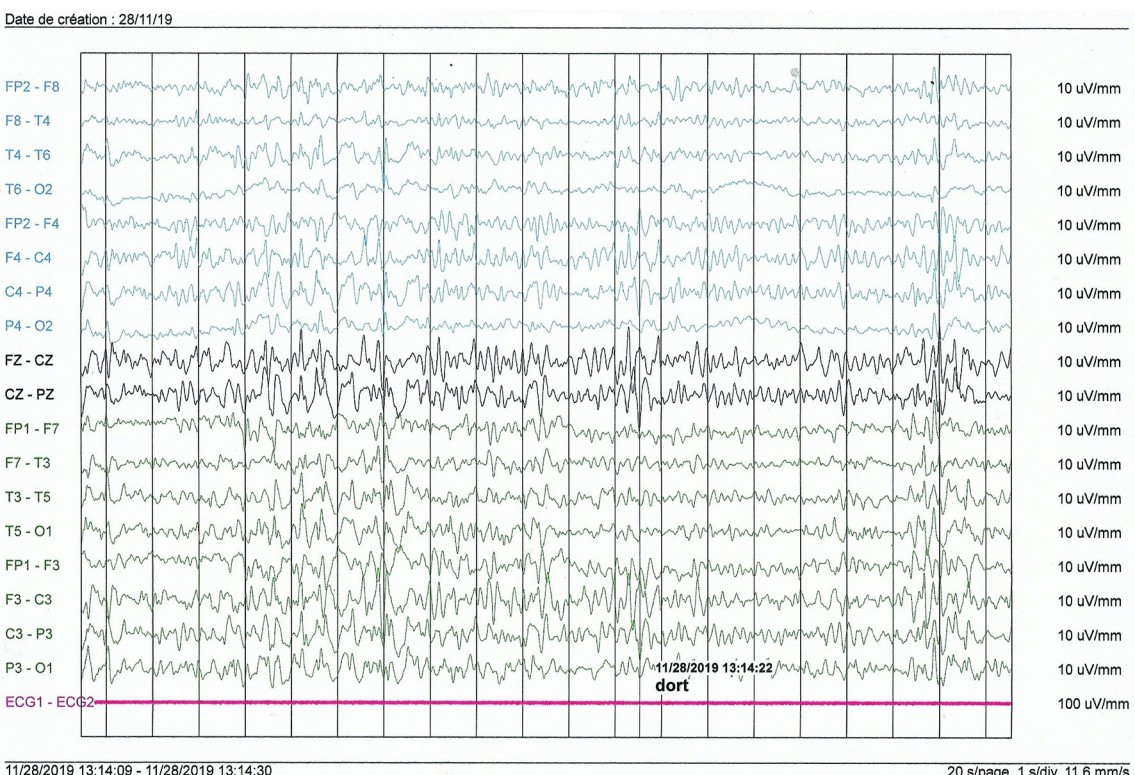

**Fig 2. Interictal electroencephalogram of an 11-year-old girl (case 1).**

Deliveries went well in all our cases and all babies cried immediately at birth, which is in agreement with other studies [6,26]. Parents did not associate HN seizures with epilepsy, which explains the lack of prior use of AED.

**Winkler's classification, HN frequency and triggers.** Two patients were classified "HN plus" according to the classification by Winkler et al. (2008). The associated epilepsy in each case was generalized epilepsy (one was myoclonic) as in most of the studies describing "HN plus" in Uganda [11,13,26] and in Tanzania [14] with variable proportions. In addition, these HN-associated seizures occurred earlier in our study compared to what was observed elsewhere [14,27].

The frequency of HN appeared to be lower in our population than in most studies conducted in East Africa. Higher frequencies were reported in some studies [18] and this is probably why the WHO refers to 5 to 20 HN/min in its definition of probable cases. However, lower frequencies were also observed elsewhere, notably in Cameroon [6]. The frequency of these seizures could be related to the severity of the disease. This is a difficult criterion to assess [6,26] and therefore often omitted [10].

In our study, as in several others, either a meal [11,13,18], or cold weather or a cold breeze [6,11,13] was the trigger for HN. In one of our cases, the mother insisted that HN were triggered when the child "ate his meal at cold hours". Seizures were triggered mainly during morning and evening meals.

**Growth retardation and sexual development.** One patient (20%) was small for their age compared to 40.9% (9/22) and 26.7% (4/15) in two series in Uganda [11,26]. It is a fairly common symptom and found in almost every series [6,18]. Physical deformities including limbs

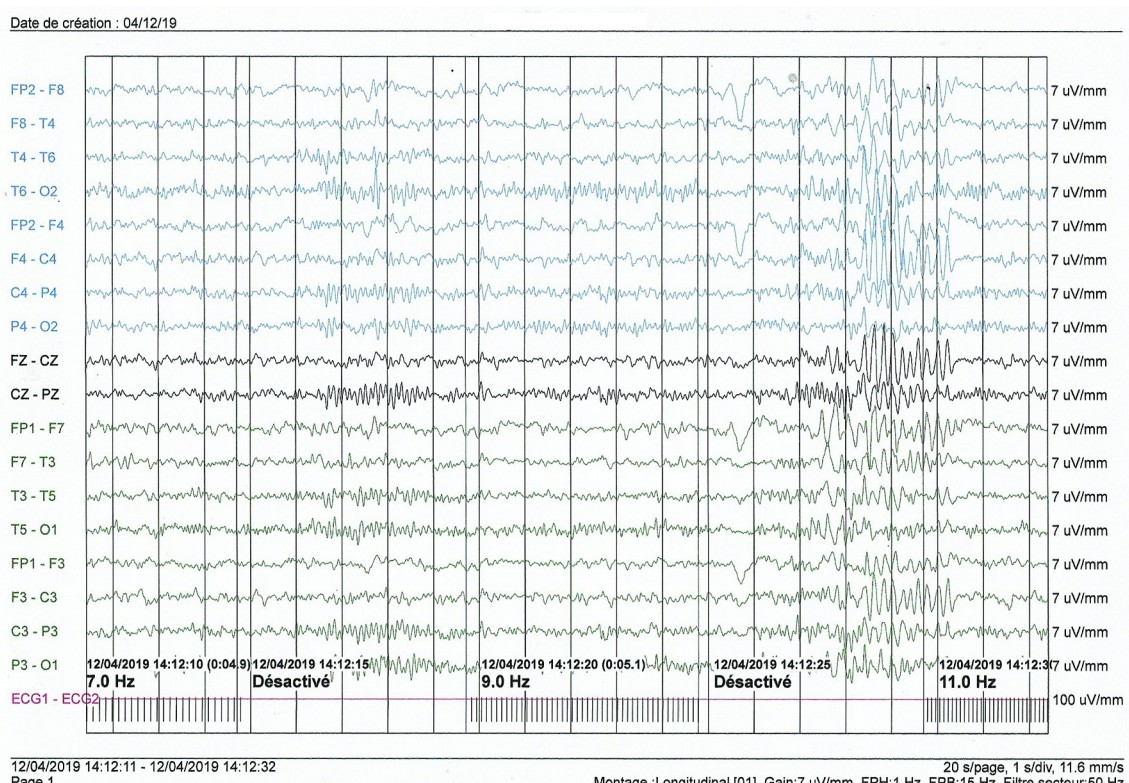

**Fig 3. Interictal electroencephalogram of an 8-year-old girl (case 4).**

that can be significant were described in some cases in our series and were responsible for gait disorders and functional impairment. These deformities were reported in somewhat older NS cases in Uganda [27].

No cases of delayed sexual development were identified among our NS cases. This would also be a sign encountered in variable proportions in the literature [2,11,18,26].

**Developmental delay and cognitive decline, psychiatric disorders.** Developmental delay and cognitive decline described in the literature [6,11–13,26,28] were also observed in our series. Only two cases (40%) showed no cognitive decline. As in Ugandan [27] and Tanzanian [14] studies, most of these signs occurred after the onset of HN (between 12 and 24 months) suggesting that this is a natural history of the disease. Cognitive impairments were relatively frequent in the different NS populations and the proportions varied according to location: 100% in Sudan [12], 45.4% in Uganda [11], and 40.3% in Tanzania [28]. The severity of these cognitive impairments also varied within each study. For example, although 100% of children had cognitive impairment in Sudan, it was severe in only 9.5% of these cases, compared to 45.4% in Uganda and 48% in Tanzania among children with cognitive impairment. Deficiencies were perhaps less severe in Central African countries, such as Cameroon [6] and CAR (this study), where cognitive deficiencies were mostly mild to moderate.

The presence of psychiatric signs (60% in our study) is quite variable from one study to another, ranging from 0% [26] to 50% [11] in the same country (Uganda). These signs as well as prolonged periods of depressive mood appeared about 24 months after HN, which is similar to what was described in Uganda [27].

Last but not least, the fact that some parents reported prodromal signs (somnolence, diminished attention, blank starring, etc.) during the months or years preceding HN onset aligns with the natural history of NS as proposed by Idro et al [27].

**Onchocerciasis endemicity.** Several studies reported a high number of NS cases in areas highly endemic for onchocerciasis [17,18,29]. No data on *Onchocerca volvulus* endemicity were available in this area at the time of the present study (December 2019). In January 2020, we conducted a follow-up study in the same area to assess the prevalence of onchocerciasis in children aged six to nine years, based on Ov16 antibody prevalence. Results were recently published and the overall Ov16 seroprevalence was 8.9% [21]. This means that there is an ongoing *Onchocerca volvulus* transmission in the Landja area, but the endemicity of onchocerciasis in the area is low. However, the overall prevalence of onchocerciasis antibodies was 20% in Kodjo, a village close to rapids on the Ubangui river [21]. This suggests that *Onchocerca volvulus* transmission is high and ongoing in remote villages along the Ubangui river and the burden of NS could be higher there, but the prevalence of the disease is unknown in this area where there are security concerns and problems of access.

## The cons

**Diagnostic certainty (WHO Classification).** According to the WHO classification [2], only one of our cases was confirmed and the other four were probable. As in other studies, it was difficult to confirm a case of NS. For example, only three out of 21 cases were confirmed in Sudan [12] and none in 15 cases in Uganda [26]. Probable cases could therefore be sufficient to diagnose the disease and to manage the patients. According to the WHO classification and literature data, the diagnosis of NS may be uncertain in some cases, particularly those with focal neurological signs or symptoms (case N˚ 5). However, such focal abnormalities in persons with NS were already described in some series, particularly in Uganda [11,13] and Tanzania [14] but they were very uncommon. Lastly, the grouping of cases in space (associated with other clinical criteria) seems to be a very important criterion to support the diagnosis of NS. It is important to recall that an expert consensus meeting in Gulu in 2015 had already raised several limitations and shortcomings of the WHO classification and suggested a modification [15].

**Electroencephalogram.** Among our patients, four had an abnormal EEG. In patients with NS, EEG results were abnormal in different proportions according to the studies, ranging from 60% [14] to 100% [11]. Most of these anomalous patterns in the literature were obtained in the interictal phase as was the case in our study [12–14]. In some cases the EEG trace changed during the ictal phase (i.e. during HN) [12], but this was not observed in our cases.

Apart from a diffuse slowing observed in two of our patients, the abnormalities noted on the EEG were variable. Only two patients had slow spikes. The various other graphoelements we found could be consistent with diffuse brain damage due to different pathologies. Also, we caught a HN seizure during the EEG exam but there was no change in the EEG tracing.

These results suggest that NS may have several electroencephalographic manifestations in this population that may correspond either to a wide variety of clinical forms or to various stages of disease progression. In addition, it is important to note that there is still no consensus of what a "typical" EEG result might be in a patient with NS.

**Limitations of the study.** The aim of this study was to describe the clinical features of NS cases in CAR but a few limitations exist. Because of the difficulty of access to certain areas, we only surveyed one of the six districts of the Landja area and the number of NS cases identified is too low to draw definite conclusions. Therefore, it is likely that the figures presented in this study are far from the real prevalence of NS. There may also be a recall bias, especially in the

reconstruction of the history of the disease by parents or guardians, which may have impacted the timing of onset, duration of symptoms and potentially the diagnosis itself. It is also possible that we missed additional NS cases because 39% of cases detected with the screening questionnaire were not seen by the neurologist. Two main reasons were mentioned by the families: some parents said they were discouraged by their children's state of health and did not want to make any more efforts; other parents were simply absent or unavailable to take their children to the health center. To our knowledge, validated questionnaires for screening and confirmation of NS cases are not available. We therefore used non-validated questionnaires that were inspired by all the studies carried out by experts in the field. There was a variability between interviewers during the screening phase because the investigators were organized in three teams but all interviewers were trained before the study, had the same degree of education, and asked only multiple choice questions. The two confirmation questionnaires and the EEG questionnaire were administered by the same team, thus eliminating inter-interviewer variability.

## Conclusion

To the best of our knowledge, our series represents the first description of probable NS in CAR. Although there may be some arguments against the diagnosis of NS in some of the cases, there seem to be more arguments that strongly suggest that the five cases described in our study should all be classified as NS. Given that only a small part of the affected area was investigated, the study area along the Ubangui river needs to be expanded in order to investigate the association between *Onchocerca volvulus* and NS and also evaluate the real burden of NS in CAR. NS is a public health problem in affected areas, but the disease is not yet known in CAR and therefore not managed efficiently. It is therefore important to sound the alarm at the level of the local health authorities in order to implement adequate care for these children.

## Supporting information

**S1 File. Questionnaire for the screening of epilepsy and nodding syndrome cases in the Central African Republic.**
(PDF)

**S2 File. Questionnaire for the confirmation of epilepsy cases in the Central African Republic.**
(PDF)

**S3 File. Questionnaire for the confirmation of Nodding syndrome cases in the Central African Republic.**
(PDF)

**S4 File. Questionnaire for the disease history of nodding syndrome cases in the Central African Republic.**
(PDF)

**S5 File. Description of the two suspected cases of NS that were excluded from the final sample.**
(PDF)

## Acknowledgments

We are very grateful to the medical students at the "Hôpital de l'Amitié" in Bangui, the Landja resident surveyors, the staff of the health center where our team was based, and the traditional

authorities for their contribution to the smooth running of this study. We would also like to thank all the patients and their relatives who gave their time to participate in this study. Many thanks to the members of the NGO Fracarita who work in the village and thanks to whom the alert was launched.

## Author Contributions

**Conceptualization:** Salvatore Metanmo, Farid Boumédiène, Pierre-Marie Preux, Robert Colebunders, Pascal Mbelesso, Daniel Ajzenberg.

**Data curation:** Salvatore Metanmo, Farid Boumédiène, Daniel Ajzenberg.

**Formal analysis:** Salvatore Metanmo, Andrea S. Winkler, Pascal Mbelesso, Daniel Ajzenberg.

**Investigation:** Salvatore Metanmo, Emmanuel Yangatimbi, Pascal Mbelesso.

**Methodology:** Salvatore Metanmo, Farid Boumédiène, Pierre-Marie Preux, Pascal Mbelesso, Daniel Ajzenberg.

**Project administration:** Farid Boumédiène, Pierre-Marie Preux, Pascal Mbelesso, Daniel Ajzenberg.

**Writing – original draft:** Salvatore Metanmo, Daniel Ajzenberg.

**Writing – review & editing:** Salvatore Metanmo, Farid Boumédiène, Pierre-Marie Preux, Robert Colebunders, Joseph N. Siewe Fodjo, Eric de Smet, Emmanuel Yangatimbi, Andrea S. Winkler, Pascal Mbelesso, Daniel Ajzenberg.

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
