## [Decision Letter · Decision Letter 0]

5 Jan 2021

Dear Pr Ajzenberg,

Thank you very much for submitting your manuscript "First description of the Nodding Syndrome in the Central African Republic" for consideration at PLOS Neglected Tropical Diseases. As with all papers reviewed by the journal, your manuscript was reviewed by members of the editorial board and by several independent reviewers. In light of the reviews (below this email), we would like to invite the resubmission of a significantly-revised version that takes into account the reviewers' comments. 

We cannot make any decision about publication until we have seen the revised manuscript and your response to the reviewers' comments. Your revised manuscript is also likely to be sent to reviewers for further evaluation.

Sincerely,

Prof. María-Gloria Basáñez, PhD, MSc

Associate Editor

Robert Reiner

Deputy Editor

Reviewer's Responses to Questions

**Key Review Criteria Required for Acceptance?**

**Methods**

-Are the objectives of the study clearly articulated with a clear testable hypothesis stated?

-Is the study design appropriate to address the stated objectives?

-Is the population clearly described and appropriate for the hypothesis being tested?

-Is the sample size sufficient to ensure adequate power to address the hypothesis being tested?

-Were correct statistical analysis used to support conclusions?

-Are there concerns about ethical or regulatory requirements being met?

Reviewer #1: The region studied was a convenience sample in the sense of accessibility, but within the population studied this was a household survey and well described. Screening was done using a series of screening questionnaires and finally a review by a neurologist to decide the definitive diagnosis. Copies of the questionnaires are in the supplementary material. For what is effectively a preliminary study, this was adequate. To draw definitive conclusions it would be best to expand the surveyed area but as a first study this is adequate. Ethical requirements were met.

Reviewer #2: The manuscript fulfills all of the requirements for Methods as listed above.

Reviewer #3: The paper by Metanmo et al. presents the investigation of epileptics in the Central African Republic to describe cases of Nodding Syndrome. Their paper reports an extremely low prevalence of NS, but still with 5 cases described. Probably because of the very low number of cases, the authors then presented the paper in a more clinical manner, describing the history of the disease and the clinical characteristics of the patients.

The authors conducted a cross-sectional community-based study in one village (Landja); and applied a door-to-door approach. The authors should clarify if they surveyed systematically all the households in the selected villages. If this was the case, the study will provide an exhaustive census of the epileptic subjects and NS cases. The authors just say “each household” (line 134), but to be precise, the authors should add in the Methods the exact number of households targeted (from a census, for example) with the number of inhabitants targeted. 

The strength of the paper is that the study has been able to carry out all stages of examinations, including examinations by neurologists and the performance of EEGs. Moreover, the authors have presented well the limitations of the study, especially regarding the problems of classifying the disease.

The clinical descriptions are complete, well presented and interesting.

Please, the authors should add information regarding the “active onchocerciasis transmission” status in the study setting (line 116), and add a specific point in the Discussion on the fact that they did not find more NS cases in this seemingly active onchocerciasis transmission setting. Information on nodule prevalence would be useful. In other words (lines 133-134): how do the authors explain that there are no data on Onchocerca volvulus endemicity when they state that this is an active onchocerciasis transmission setting?

Lines 105-106: Please, add a reference to substantiate this sentence, because it is important to highlight previous reports on possible NS cases in CAR.

Mental retardation could be more explained.

Please see below: ‘Mental retardation’ has largely been supplanted with a less stigmatizing term, such as ‘developmental and cognitive regression’. Please use the latter term instead of Mental retardation.

**Results**

-Does the analysis presented match the analysis plan?

-Are the results clearly and completely presented?

-Are the figures (Tables, Images) of sufficient quality for clarity?

Reviewer #1: Yes. Results are well presented and the tables complete and informative.

Reviewer #2: The Results section analysis plan and the analysis presentation match. The tables and figures are adequate.

Reviewer #3: The results are well presented and correspond to the proposed Methods. They are, however, essentially clinical descriptions, not based on statistical evaluations.

However, the authors should present the percentage of the population evaluated in relation to the target population (see problems in the Methods highlighted above).

Line 186: 5 NS cases among 55 PWE: 9.1% and not 8.3%? please clarify this percentage, or correct it if there is a mistake. If this is a mistake, please, check all your values throughout the manuscript.

Although the authors focused on NS, results on retarded growth and mental retardation may be included for all this population.

Please see below: ‘Mental retardation’ has largely been supplanted with a less stigmatizing term, such as ‘developmental and cognitive regression’. Please use the latter term instead of Mental retardation.

**Conclusions**

-Are the conclusions supported by the data presented?

-Are the limitations of analysis clearly described?

-Do the authors discuss how these data can be helpful to advance our understanding of the topic under study?

-Is public health relevance addressed?

Reviewer #1: Because only 5 probable cases were found, numbers are not high but considering this is an area that has not been described before, the conclusions are valid within the limits of the survey. This is of considerable public health importance although the pathogenesis of nodding syndrome is still poorly understood. The conclusion that the area needs more study is certainly valid and from the public health point of view, the area needs an epidemiological evaluation of onchocerciasis urgently which could have been mentioned in the text.

Reviewer #2: The conclusions are supported by the data. The limitations, while clearly described, are insufficient and require some additional work. There is no comment on public health relevance or an extrapolation on how the data can be useful in our overall understanding of the topic.

Reviewer #3: (No Response)

**Editorial and Data Presentation Modifications?**

Reviewer #1: General Comments to the authors:

Although NS has not been described before in CAR, epilepsy has been. It would be good to include a reference to this in the background information.

Specific Comments to authors:

Line 45: We identified 5 "probable" cases of which one was was clarified NS according to the WHO definitions. This is what the survey showed but it is a bit confusing as written in the Abstract.

Line 63: i would add "since its first "described" appearance 60 years ago

Line 65: "and" psychiatric

Line 165: How do the CDC figures relate to the CAR population?

Line 181: The seven cases identified were based on what level of survey. Two were excluded later. Who made these decisions based on what?

Line 221: started "in" 2019

Line 315: Physical deformities of the limbs needs more description. My impression from the text is that these were functional deformities of the lower limbs, not physical ones.

Line 330: Is this just Central Africa or the Central African Republic? Needs correction.

Line 354: Does the 80% refer to all patients with epilepsy or just NS? 80% of a sample of 5 is stretching percentages a little!

Line 372: Could be an overestimation as well as an underestimation.

Line 382: first description of "probable" NS in CAR

Reviewer #2: Minor Revision

Reviewer #3: - Check for missing spaces (as in line 341 with reference 2)

- Clarify affiliation 3: which unit of the IRD?

- Harmonise references: e.g. ref 4 (the month is not indicated), and ref 5 (the month of publication is indicated). Please use PLoS NTDs guidelines for preparation of References.

**Summary and General Comments**

Reviewer #1: This was an important survey and establishes the probability of NS in CAR as well as epilepsy. 

It definitely needs to be extended to the larger area. I would like to have seen some information on onchocerciasis prevalence in the paper. With the REMO mapping this was not considered an area of importance, perhaps because at that time it was not populated. Were these old well established communities or were they new communities?

Reviewer #2: PNTD-D-20-02016

First Description of Nodding Syndrome in the Central African Republic

Comments to Authors:

In this manuscript, the authors report an observational cross-sectional descriptive study of a clinical syndrome consistent with Nodding Syndrome (NS) in the Central African Republic (CAR). They describe the results of a community-based survey, during which they identify a number of persons with epilepsy, and identify a small subset of these persons whom, they state, suffer signs and symptoms consistent with NS. This would represent the first description of NS in CAR, and one of the only descriptions of the syndrome outside of East Africa.

Given that the substantive evidence that what the authors are observing / describing is in fact NS, the authors are wise to list Pros and Cons arguing for and against NS in this manuscript. Only one of the 5 described cases was ‘confirmed’ per WHO case classification criteria, while the other 4 described were considered ‘probable’. Although this does speak of the difficulty in documenting head nodding by witness in these cases, and the challenges of ‘confirming’ an event that is intermittent and paroxysmal, nonetheless it would be more reassuring and convincing if there were more observations of the head nodding behavior, to ensure that the behavior noted was in fact NS, and not a NS ‘mimic’ such as partial complex or generalized atonic seizures.

Specific Comments and Questions:

1. Lines 103-105 – ‘…the only significant association was found in certain areas with onchocerciasis or river blindness…’. This statement is not entirely true. Investigations have found a small, but significant association with being seropositive for Onchocerca infection. And, only in limited areas (e.g., there is plenty of onchocerciasis in areas in which NS, or for that matter higher incidence of epilepsy, is not found). The focus on onchocerciasis, in the absence of any data in the manuscript about the situation of onchocerciasis in this area of CAR (e.g., whether it is endemic / hyperendemic; the proportion of persons infected with Onchocerca volvulus) is another attempt by some of the authors on the manuscript to push forth their hypothesis on the relationship between O. volvulus and epilepsy in general, and NS specifically.

2. Lines 137-138 – the authors state that the initial screening questionnaire was piloted in healthy volunteers…but without performance validation’. This is disturbing. The authors screened their population in order to get to the lowest denominator using an instrument that was not assessed for its sensitivity and specificity to identify epilepsy / seizures. This deserves a comment.

3. The use of no fewer than 4 individual questionnaires is worth comment as well. How did the investigators ensure minimum inter-observer /inter-evaluator variability? How were all these forms piloted?

4. Line 166 and throughout – ‘Mental retardation’ has largely been supplanted with a less stigmatizing term, such as ‘developmental and cognitive regression’. Please use the latter term instead of Mental retardation.

5. Results, Line 179 – I’m a bit confused about the population being assessed; the total interviewed was 6,175 out of a total population of over 35,000? More detail needs to be provided as to how households were chosen / persons enrolled.

6. I’m unclear why the child with severe acute malnutrition and HN was excluded from assessment. Is it not likely that many of these children, perhaps also those demonstrating HN, also suffered from severe malnutrition?

7. Case 1 – if the child was witnessed to have HN only for 30 seconds, I don’t know how reliably this finding could be extrapolated out to 1 full minute…

8. It’s a bit concerning that the one EEG that was conducted during an actual nodding episode was without electrographic correlate…

9. While the WHO criteria are challenging to meet for confirmed cases, this is what was intended when the criteria were developed. Particularly in an area in which an attempt is being made to substantiate whether NS occurs / exists in the area, it is important to directly observe the head nodding episodes, and ideally in more than one subject. Once head nodding is substantiated to be occurring, then the criteria can be loosened a bit.

10. The fact that nearly 40% of the cases identified in the screening process were not evaluated by a neurologist is also concerning, and suggests that there might have been an insufficient number of patients with head nodding to differentiate other forms of seizures from ‘head nodding’.

Reviewer #3: In conclusion, this MS could form the basis for a possible publication. But is this really the first time that cases of NS are reported in CAR? Published perhaps, but reported? and the paper requires further discussion on the very low number of NS while the authors indicate that it is an active transmission area for onchocerciasis.

PLOS authors have the option to publish the peer review history of their article (what does this mean?). If published, this will include your full peer review and any attached files.

Reviewer #1: No

Reviewer #2: No

Reviewer #3: No
---

## [Decision Letter · Decision Letter 1]

30 Apr 2021

Dear Pr Ajzenberg,

We are pleased to inform you that your manuscript 'First description of Nodding Syndrome in the Central African Republic' has been provisionally accepted for publication in PLOS Neglected Tropical Diseases.

Best regards,

María-Gloria Basáñez, PhD, MSc

Associate Editor

Robert Reiner

Deputy Editor

Reviewer's Responses to Questions

**Key Review Criteria Required for Acceptance?**

**Methods**

-Are the objectives of the study clearly articulated with a clear testable hypothesis stated?

-Is the study design appropriate to address the stated objectives?

-Is the population clearly described and appropriate for the hypothesis being tested?

-Is the sample size sufficient to ensure adequate power to address the hypothesis being tested?

-Were correct statistical analysis used to support conclusions?

-Are there concerns about ethical or regulatory requirements being met?

Reviewer #1: -Are the objectives of the study clearly articulated with a clear testable hypothesis stated? YES: THIS IN THE END WAS A SIMPLE SURVEY WITH CLINICAL PRESENTATIONS BUT HAS SHOWN THE POSSIBLE PRESENCE OF NS

-Is the study design appropriate to address the stated objectives? YES. ALL THAT COULD BE DONE IN THE CIRCUMSTANCES

-Is the population clearly described and appropriate for the hypothesis being tested? YES AS AN INITIAL SURVEY

-Is the sample size sufficient to ensure adequate power to address the hypothesis being tested? FOR A PRELIMINARY STUDY OK BUT AS INDICATED A LARGER STUDY IS NECESSARY TO DRAW DEFINITIVE CONCLUSIONS.

-Were correct statistical analysis used to support conclusions? ONLY SIMPLE % WERE USED DUE TO SMALL NUMBERS

-Are there concerns about ethical or regulatory requirements being met? NO

Reviewer #3: (No Response)

**Results**

-Does the analysis presented match the analysis plan?

-Are the results clearly and completely presented?

-Are the figures (Tables, Images) of sufficient quality for clarity?

Reviewer #1: RESULTS ARE CLEARLY PRESENTED AND THE TABLES ARE ADEQUATE AND CLEAR.

Reviewer #3: (No Response)

**Conclusions**

-Are the conclusions supported by the data presented?

-Are the limitations of analysis clearly described?

-Do the authors discuss how these data can be helpful to advance our understanding of the topic under study?

-Is public health relevance addressed?

Reviewer #1: THERE IS VERY LITTLE DATA AND BASICALLY THIS HAS BECOME A CASE PRESENTATION. THE AUTHORS HAVE REALISED THIS. I THINK IT WOULD BE BETTER TO PUT A "PROBABLE CASE" OF NS IN THE TITLE OF THE PAPER

Reviewer #3: (No Response)

**Editorial and Data Presentation Modifications?**

Reviewer #1: GeneraL comments

1. CAR has had a national programme for onchocerciasis for many years and REMO was carried out under APOC. Why was this area not treated? Was it considered hypo endemic or uninhabited? It would be good to have some information on this. Can the authors add a REMO map for this part of CAR?

2. Only one case fulfilled all the criteria for NS. Onchocerciasis transmission is ongoing but not that high. All the EEG studies showed different patterns. Another weakness of this paper is the small numbers but it could be strengthened with more onchocerciasis data. Some was collected after the original study and this is helpful but it woUld be good to know the status of the patients and the entourage.

3. Although this was a study of NS, the association of onchocerciasis and epilepsy is very strong, even if no causal link is established. It would add weight to the argument that this could be related to onchocerciasis if there were some epilepsy statistics from the same area.

4 Some work has been done on epilepsy in the CAR by one of the authors even if not a strong paper. A reference to epilepsy in the CAR would add some strength to the paper.

Some minor text corrections

Line 35. The disease has been limited to East Africa, actually it has only been described in East Africa. It would be more correct to say "The description of the disease has been limited to East Africa"

Line 69 -70 Because the risk of NS was only partially investigated in the study we "strongly recommend" a larger scale....

Line 191 Female aged 11 years who has never been to school

Line 195 memory deficits

Line 296 The births of all the cases were normal

Line 322 Secondary sexual characteristics would not be expected in two of these children anyway due to age!

Reviewer #3: (No Response)

**Summary and General Comments**

Reviewer #1: This paper describes some field work done under difficult conditions and has highlighted the need for a more systematic approach to studies of the transmission of onchocerciasis, the presence of NS and the possibility of epilepsy which also should be investigated. A further study in a wider area with good statistics would be very useful.

Reviewer #3: Metanmo and colleagues accurately answered all the comments; and were able to supplement the informations, particularly on onchocerciasis endemicity (both in the introduction and discussion), with information on anecdotal cases raised by Central African physicians. The importance of screening in terms of number of populations examined, the systematic nature of the examinations with neurological and EEG evaluations make this paper an important and informative contribution. The conclusion brought is essentially to push for wider investigations and towards areas of suspected onchocerciasis to be higher endemic, which is consistent with the results. I therefore recommend this paper for publication.

Two minor elements to modify for the paper: 1) the affiliation does not seem correct for IRD, but rather seems to be Associated Unit number 997 NET; and 2) please in the references, use italic terms when necessary (see e.g. ref 8).

PLOS authors have the option to publish the peer review history of their article (what does this mean?). If published, this will include your full peer review and any attached files.

Reviewer #1: No

Reviewer #3: No

---

## [Editor Report · Acceptance letter]

14 Jun 2021

Dear Pr Ajzenberg,

We are delighted to inform you that your manuscript, "First description of Nodding Syndrome in the Central African Republic," has been formally accepted for publication in PLOS Neglected Tropical Diseases.

Best regards,

Shaden Kamhawi

co-Editor-in-Chief

Paul Brindley

co-Editor-in-Chief
